# Complete Chloroplast Genome Sequence of *Dahlia imperialis* (Asteraceae): Comparative Analysis and Phylogenetic Relationships

**Shan-De Duan** [1], **Yang Liu** [1], **Li-Hong Hao** [1,*], **Di-Ying Xiang** [1], **Wen-Bin Yu** [2], **Juan Liang** [3], **Duan-Fen Chen** [1] and **Shan-Ce Niu** [1,4,*]

1   College of Horticulture, Agricultural University of Hebei, Baoding 071001, China; duansd940603@163.com (S.-D.D.); lyang15502416083@163.com (Y.L.); yyxdy@hebau.edu.cn (D.-Y.X.); chenduanfen@163.com (D.-F.C.)
2   Zhangjiakou Academy of Agricultural Sciences, Zhangjiakou 075000, China
3   Shijiazhuang Botanical Garden, Shijiazhuang 050073, China
4   State Key Laboratory of Crop Improvement and Regulation in North China, Agricultural University of Hebei, Baoding 071001, China
*   Correspondence: haolihong1986@163.com (L.-H.H.); niushance@163.com (S.-C.N.)

**Abstract:** The genus *Dahlia* has approximately 40 species; however, the complete chloroplast genome has been reported only for one species. Due to this lack of information on the chloroplast genomes, the phylogenetic relationships within the *Dahlia* genus remain unclear. Therefore, the present study sequenced the complete chloroplast genome of *D. imperialis* for the first time. This genome was 152,084 bp long with a typical quadripartite structure and a GC content of 38.45%. A total of 134 genes were annotated in the genome, including 86 protein-coding genes, 38 transfer RNA genes, 8 ribosomal RNA genes, and 2 pseudogenes. The detailed analysis identified UUA as the most frequently used codon and found 62 intergenic repeat sequences and 157 SSR loci in the *D. imperialis* genome. Phylogenetic analysis based on 49 chloroplast genomes showed that *Dahlia* was a monophyletic group, with *D. imperialis* positioned at the base of this clade. Network evolution and gene flow analysis unveiled extensive hybridization events within the Heliantheae alliance, especially in *Dahlia*. Thus, the comprehensive analysis of the complete chloroplast genome of *D. imperialis* enriches the information on the genetic resources of the Dahlia species, provides valuable information for reconstructing their phylogeny, and reveals the evolutionary dynamics of the *Dahlia* chloroplast genome.

**Keywords:** Asteraceae; Heliantheae alliance; chloroplast genome; *Dahlia imperialis*; phylogeny





## 1. Introduction

*Dahlia*, also known as "sweet potato flower" and "Oriental chrysanthemum", is a perennial bulbous flower of the Heliantheae alliance in the Asteraceae family. It is native to Mexico and is the country's national flower [1,2]. In the early 16th century, it was introduced to the courtyards by locals, and in the 17th century, it was brought to Europe. Currently, there are more than 30,000 cultivated *Dahlia* varieties, making it one of the species with the most flower varieties in the world [3]. *Dahlia* is renowned for its prolonged flowering period and large and numerous flowers with diverse colors and shapes. Moreover, its tubers are rich in inulin. *Dahlia* is a multi-functional horticultural plant with ornamental, medicinal, and edible value [4–7]. In recent years, *Dahlia* has become increasingly popular as a cut flower, in addition to its use as a traditional potted plant and in flower arrangements. *Dahlia*'s cut and potted flower industries occupy a significant position in the floriculture economy of various countries; it is also one of the most important economic groups in the Heliantheae alliance (Asteraceae) [3]. *Dahlia*'s cultivation and domestication history can be traced back a few hundred years. However, research on the *Dahlia* genus mainly focused on functional components and breeding, especially in *D. pinnata* [8,9]. Therefore, other species of *Dahlia* have not been fully studied. In addition, there are approximately 40 species in the

*Dahlia* genus [10], but existing horticultural cultivars are only derived from two distinctive species from Mexico: *D. pinnata* and *D. coccinia* [1,2,4]. These have since developed into an ornamental plant that is popular worldwide, with over 3000 decorative varieties. A significant number of genetic resources in the *Dahlia* genus remain underutilized.

The Heliantheae alliance, with more than 6600 species, is the largest tribe in the Asteraceae family. It comprises multiple subtribes and is found in all continents except Antarctica. However, more than 85% of these species are in South and North America [11]. The high species diversity and extensive distribution of the Heliantheae alliance have posed significant challenges for studying their phylogenetic relationships. Moreover, research has demonstrated inconsistencies in phylogenetic relationships within the Heliantheae alliance [12]. In the Asteraceae family, phylogenetic trees constructed based on nuclear genomes [13] and chloroplast gene loci [11] demonstrated apparent inconsistencies. Although the basic phylogenetic framework of Asteraceae species has been established, further research is needed to understand the phylogenetics of *Dahlia* with other subtribes and genera of the alliance.

Chloroplasts are semi-autonomous cell organelles in plant cells and have a genome composed of multiple copies of a circular DNA fragment (110–210 kb) [14]. The chloroplast genome is generally small and easy to assemble, has a low recombination rate, and is amenable to analysis. Most plant chloroplast DNAs have a typical quadripartite structure [15], including a large single-copy region (LSC), a small single-copy region (SSC), and a pair of inverted repeats (IRs). The chloroplast genome of most land plants contains 110–130 genes [16]. These chloroplast genes belong to the maternal lineage, and using them to construct phylogenetic trees helps avoid interference from paternal lineage genes. Moreover, compared to the nuclear genome, the chloroplast genome has evolved slowly and possesses higher sequence conservation [17], a major advantage in tree-building. Thus, the chloroplast genome is used in plant classification and species-level phylogenetics [18], such as *Rheum* (Polygonaceae) [19], *Paphiopedilum* (Orchidaceae) [20], etc. In addition, the phylogeny of the *Dahlia* genus, with approximately 40 species [1], remains unclear [10]. Only the chloroplast genome of *D. pinnata* has been reported, and it is crucial to generate the complete chloroplast genomes of other *Dahlia* species to elucidate the phylogeny.

Therefore, the present study aimed to enrich the information on the genetic resources of the *Dahlia* species and to determine the phylogenetic position of the *Dahlia* genus and the complex phylogenetic relationships among species within the Heliantheae alliance. We sequenced, assembled, and annotated the chloroplast genome of *D. imperialis* and systematically analyzed its basic structure and gene sequences. We further downloaded the complete chloroplast genomes of 46 Heliantheae alliance species and three outgroups from NCBI and reconstructed the phylogenetic relationships within the Heliantheae alliance. By studying the relationships of inconsistent phylogenetic groups within the Heliantheae alliance and conducting reticulate evolution and gene flow analysis, the study's findings will improve our understanding of the phylogenetic relationships within the Heliantheae alliance and the evolution of Heliantheae alliance species. The interpretation of chloroplast genome information of *D. imperialis* will enrich the genetic information we have on *Dahlia* and will lay a foundation for the innovation and application of *Dahlia* germplasm resources.

## 2. Materials and Methods

### 2.1. Plant Materials

Fresh leaves of *D. imperialis* were collected from the greenhouse test park (38°82′ N, 115°44′ E) located within the east campus of Hebei Agricultural University in Baoding City, Hebei Province. One such specimen has been deposited at the College of Horticulture, Hebei Agricultural University (https://yuanyi.hebau.edu.cn/, accessed on 11 September 2021) under voucher number DLH20210911. Fresh leaves were disinfected with 75% alcohol, then rinsed with water 3 times and stored at −80 °C after liquid nitrogen precooling.

## 2.2. DNA Extraction, Genome Sequencing, and Annotation

Whole genomic DNA was extracted from fresh leaves using a rapid plant genomic DNA isolation kit (Sangon Biotech Co., Ltd., Shanghai, China). The quality of DNA was checked using a BioPhotometer Plus (nucleic acid protein detector, Eppendorf, Germany) and 1% agarose gels. The genomic DNA was physically disrupted (by ultrasonic oscillation) into fragments 350 bp in size. Then, the small fragment sequencing library was constructed with a TrueLib DNA Library Rapid Prep Kit for Illumina (Ikesai Biotechnology Co., Ltd., Shanghai, China). The genome sequencing was implemented utilizing the Illumina NovaSeq 6000 platform, and it generated approximately 10 Gb of raw data. The quality of the raw paired-end reads was assessed using FastQC v0.11.9 [21] software. The reads were assembled using GetOrganelle [22], and the assembled genome was checked using Burrows–Wheeler Aligner (BWA) [23] and SAMtools [24]. The assembled genome's maximum and minimum read mapping depths were $1538\times$ and $83\times$, respectively (Figure S1). Further, the genome was annotated using the online annotation tool CPGAVAS2 (http://47.96.249.172:16019/analyzer/annotate, accessed on 11 September 2023) [25] with *Dahlia* as a reference, and the annotation results were manually corrected. Finally, the assembled chloroplast genome was visualized using CPGview [26]. The annotated genome has been submitted to the NCBI database under GenBank accession number OP323060.1.

## 2.3. Codon Usage Analysis

The CDS was analyzed based on relative synonymous codon usage (RSCU) using CodonW 1.4.2 software [27]. The average of three bases (GC_all) and the GC content of the first codon base ($GC_1$), second codon base ($GC_2$), and third codon base ($GC_3$), as well as the frequency of G or C in the third base ($GC_3S$), were determined using the online program CUSP (http://itime.med.ucm.es/EMBOSS/, accessed on 11 September 2023), whereas the ENC (effective number of codons) values were generated using the CHIPS model (http://itime.med.ucm.es/EMBOSS/, accessed on 11 September 2023). We then analyzed the content of A, T, C, and G at the third position of each codon based on the parity rule 2 (PR2)-bias plot, generated using $A_3/(A_3 + T_3)$ as the ordinate and $G_3/(G_3 + C_3)$ as the abscissa. Each base composition was displayed in a plane, and the center point represented the codon state under unbiased usage; here, A = T and C = G. In addition, the vector distance between other points and the center point represented the degree and direction of bias [28]. Further, a neutrality plot was created, representing the $GC_3$ value along the *x*-axis and the $GC_{12}$ value (the average of $GC_1$ and $GC_2$ for each gene) along the *y*-axis. Each point in this plot represented an independent gene [29]. Then, the ENC plot and a two-dimensional scatter plot were generated using the $GC_3$ value along the *x*-axis and the ENC value along the *y*-axis [30].

## 2.4. Interspersed Repeats and SSRs

The repetitive sequences in the chloroplast genome of *D. imperialis* were detected using the REPuter program [31] (BiBiServ2-REPuter (uni-bielefeld.de)) with the following set parameters: (1) a Hamming distance of 3, (2) a minimum size of 30 base pairs, and (3) a sequence identity of at least 90%. The analysis identified four types of repeats (palindromic, forward, reverse, and complement repeats).

Further, the SSRs in the chloroplast genome were identified using the online tool MISA [32] (https://webblast.ipk-gatersleben.de/misa/, accessed on 11 September 2023), with the following thresholds: eight repeats for mononucleotides, five for dinucleotides, four for trinucleotides, and three for tetranucleotides, pentanucleotides, and hexanucleotides.

## 2.5. Comparison of the Complete Chloroplast Genomes

Ten chloroplast genome datasets (*Lasthenia burkei*, *Galinsoga quadriradiata*, *Tagetes erecta*, *Cosmos bipinnnatus*, *Helianthus grosseserratus*, *Bidens campylotheca*, *Eupatorium fortunei*, *D. pinnata*, *D. imperialis*, *Centipeda minima*) were selected based on the phylogeny recon-

structed using nuclear genomes [13] for comparative analysis, including those of the two *Dahlia* species for comparing sequence variation within the genus *Dahlia*. *Dahlia*, *Bidens*, and *Cosmos* species were used to compare the sequence variation in Coreopsis. In addition, one representative species was selected from the other five subtribes, along with the four Coreopsideae species and an outgroup, to compare sequence variation at the level of the Heliantheae alliance.

Then, to compare the homologous gene sequences across different plants, we employed MAFFT v7 [33]. The nucleotide diversity (Pi) value for each gene was calculated using DNAsp v5.0 [34], and the IR, SSC, and LSC boundaries were visualized using CPJS-draw v1.0.0 [35]. The Ka/Ks values of the genes were calculated using the Ka/Ks Calculator v2.0 [36] software.

### 2.6. Phylogenetic Analysis

In this study, to determine the phylogenetic position of *D. imperialis*, 46 complete chloroplast genomes of the Heliantheae alliance and 3 outgroup species, *C. minima* (NC 065155.1), *Blumea balsamifera* (BK013127.1), and *Ligularia fischeri* (NC 039352.1), were downloaded from NCBI (Table S1). All sequences were first aligned using MAFFT v7 [33]. Then, the best model, GTR + R, was determined, and the maximum likelihood analysis was performed using PhyML 3.0 [37]. The network evolution was analyzed using Splitstree v4.19.1 [38] software with default parameter settings. The gene flow was analyzed by running Treemix v1.13 [39] software for 1000 iterations, and the most likely eight instances of gene flow were selected.

## 3. Results

### 3.1. Chloroplast Genome Features

The chloroplast genome of *D. imperialis* had a typical circular quadripartite structure. The complete genome sequence was 152,084 bp long, with an LSC region of 83,679 bp, an SSC region of 18,343 bp, and IRa/b regions of 25,031 bp. The total GC content of the chloroplast genome was 38.45%, and the GC distribution was uneven within the genome, with the IR region (43.01%) having a higher GC content than the LSC (35.63%) and the SSC (31.15%) regions (Table 1). The chloroplast genome of *D. imperialis* was predicted to contain 134 genes (Figure 1, Table S2), including 86 protein-coding genes, 38 tRNA genes, 8 rRNA genes, and 2 pseudogenes. Among them, 15 genes (*atpF*, *ndhA*, *ndhB*, *petB*, *petD*, *rpl2*, *rpl16*, *rps16*, *rpoC1*, *trnI-GAU*, *trnG-GCC*, *trnL-UAA*, *trnV-UAC*, *trnA-UGC*, *trnK-UUU*) had a single intron and 4 (*clpP*, *ycf3*, and 2 *rps12*) had two introns.

**Table 1.** Characteristics of *D. imperialis* chloroplast genome.

| Category | Item | Describe |
|---|---|---|
| | Cp genome/bp | 152,084 |
| | LSC/bp | 83,679 |
| Cp genome structure | SSC/bp | 18,343 |
| | IRa/IRb/bp | 25,031 |
| | Cp gene | 134 |
| | tRNA | 38 |
| Gene composition | rRNA | 8 |
| | Protein coding | 86 |
| | Pseudo | 2 |
| | Cp gene | 38.45 |
| | LSC | 35.63% |
| GC content (%) | SSC | 31.15% |
| | IRa/IRb | 43.01% |

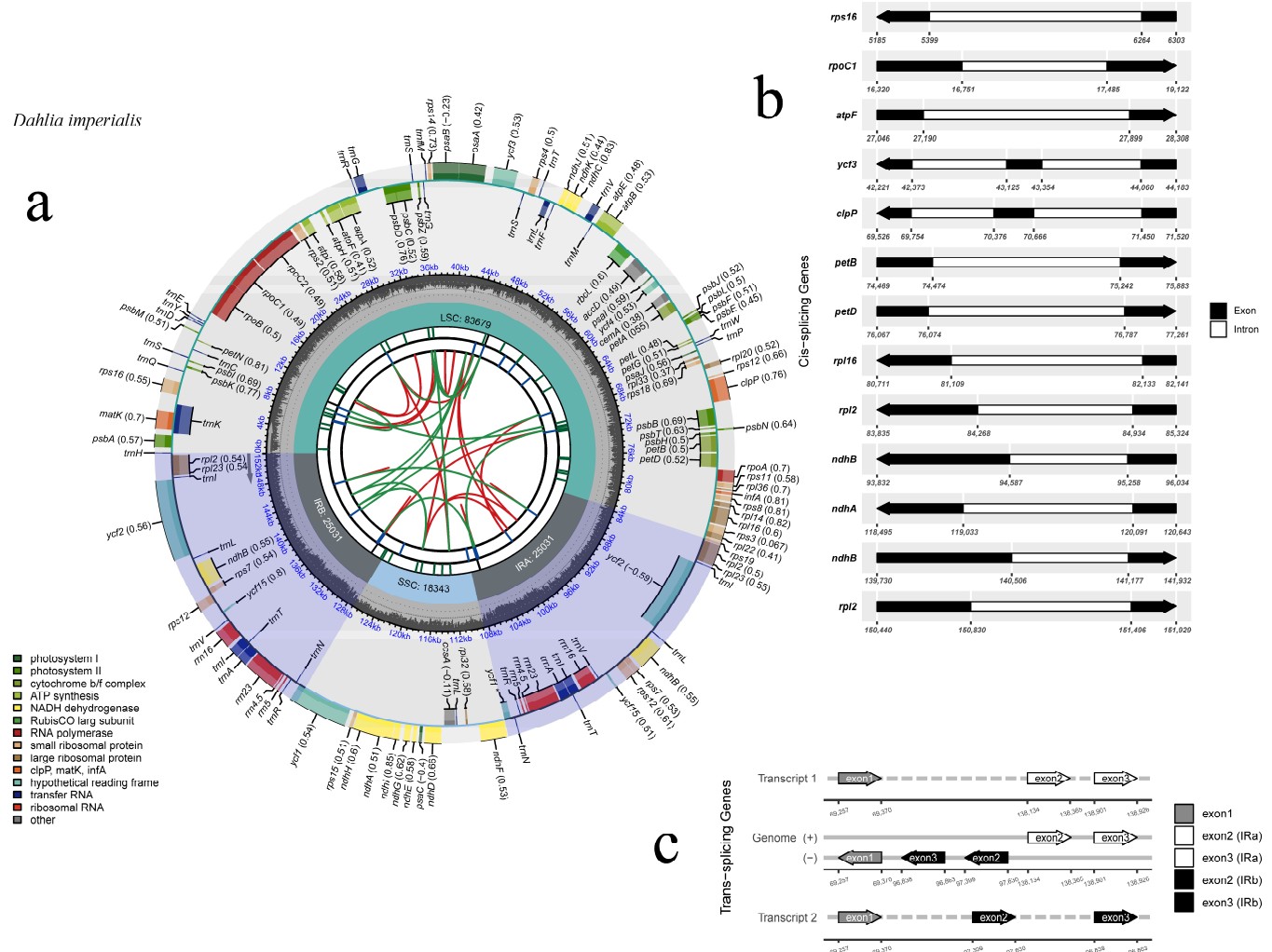

**Figure 1.** Chloroplast genomic map of *D. imperialis*. (**a**) The circle map of the complete plastome of *D. imperialis*. Genes drawn inside the circle are transcribed clockwise, whereas those outside are transcribed counterclockwise. Genes belonging to different functional groups are color coded. The dashed area in the inner circle indicates the GC content. (**b**) The black–white arrow shows the cis-spliced genes, and (**c**) the black–grey–white arrow shows the trans-spliced genes (rps12).

### 3.2. Codon Usage Bias (CUB)

A correlation analysis was performed to identify the relationship among the codon usage parameters of the *D. imperialis* chloroplast genome (Figure 2). The analysis revealed a highly significant correlation between $GC_1$ and $GC_2$ of the codons but no significant correlation between $GC_1$, $GC_2$, and $GC_3$. Both $GC_2$ and $GC_3$ showed a correlation with the codon number (condon.No.). Furthermore, a highly significant correlation was detected between $GC_3$ and the effective number of codons (ENCs), indicating a significant impact of the composition of the third base of the codon on codon preference. Meanwhile, the correlation between ENC and codon number was insignificant, suggesting a minimal effect of the sequence length on codon preference.

In addition, we analyzed the third base of the codons. Firstly, we explored the relationship between ENC and $GC_3$ distribution (Figure 3a), with most data points falling within the standard curve and showing a non-uniform distribution. Furthermore, we analyzed the distribution of the third base of the codons (Figure 3b), with most data points mainly located in the lower-right quadrant, indicating a preference in the frequency of the third base of the codons, with C having a higher frequency of use than G and A having a higher frequency of use than T. Finally, we analyzed the synonymous mutations of the third base

of degenerate codons (Figure 3c), with all data points located above the standard curve and most points distributed on both sides of the regression line. The $R^2$ value of 0.0058 indicates no significant correlation between the $GC_3$ and $GC_{12}$. In conclusion, we speculate that the codon preference of *D. imperialis* is influenced not only by natural variation but also to some extent by natural selection.

CUB describes the variation in the usage of synonymous codons in DNA [40]. CUB varies between species and between genes within a species due to a combination of factors, including mutation, selection, and drift during the long-term evolutionary process [41]. In this study, we calculated the RSCU (relative synonymous codon usage) to assess the codon usage frequencies of protein-coding genes in the *D. imperialis* genome. Here, 21,098 codons (including three stop codons) were predicted to code for all proteins (Figure 4 and Table S3). Interestingly, leucine (Leu) had the highest number of codons (2235, 10.59%), followed by isoleucine (Ile) (1795, 8.51%) and serine (Ser) (1562, 7.40%), whereas cysteine (Cys) had the least common codon (226, 1.07%). A total of 30 codons (46.8%) had an RSCU value greater than 1. The most preferred codon was TTA, which encoded Leu with an RSCU value of 1.9572, followed by AGA, which encoded arginine (Arg) with an RSCU value of 1.8534. Our results also show a significantly higher frequency of A/T-ending codons than G/C-ending codons, indicating a clear codon usage preference in *D. imperialis*.

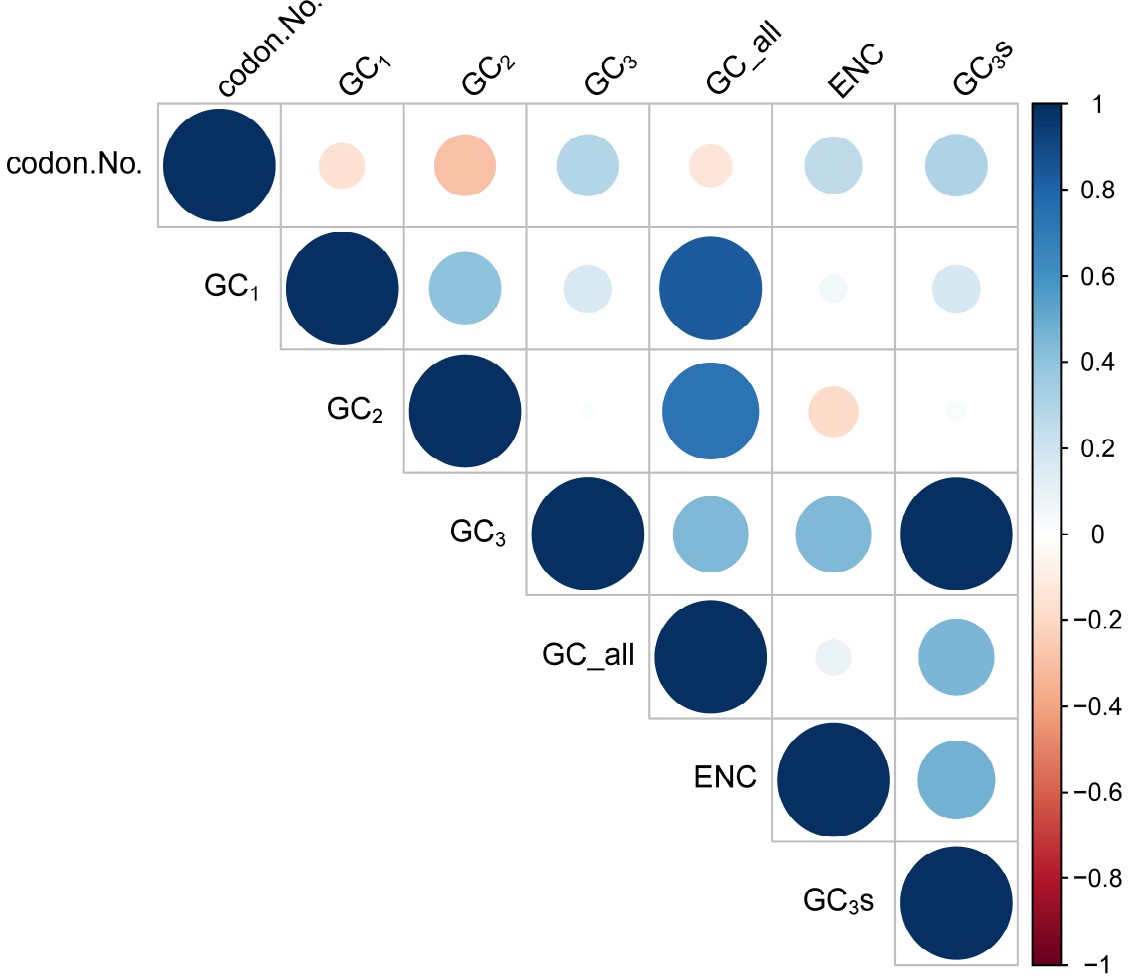

**Figure 2.** Correlation among the *D. imperialis* chloroplast genome codon usage parameters. GC_all: the average content of G or C in the three bases of the codon. $GC_1$: the G or C content in the first base of the codon. $GC_2$: the G or C content in the second base of the codon. $GC_3$: the G or C content in the third base of the codon. $GC_3$s: the frequency of G or C in the third base of the codon. ENC: effective number of codons. Codon.No.: number of codons. The larger the circle, the higher the correlation.

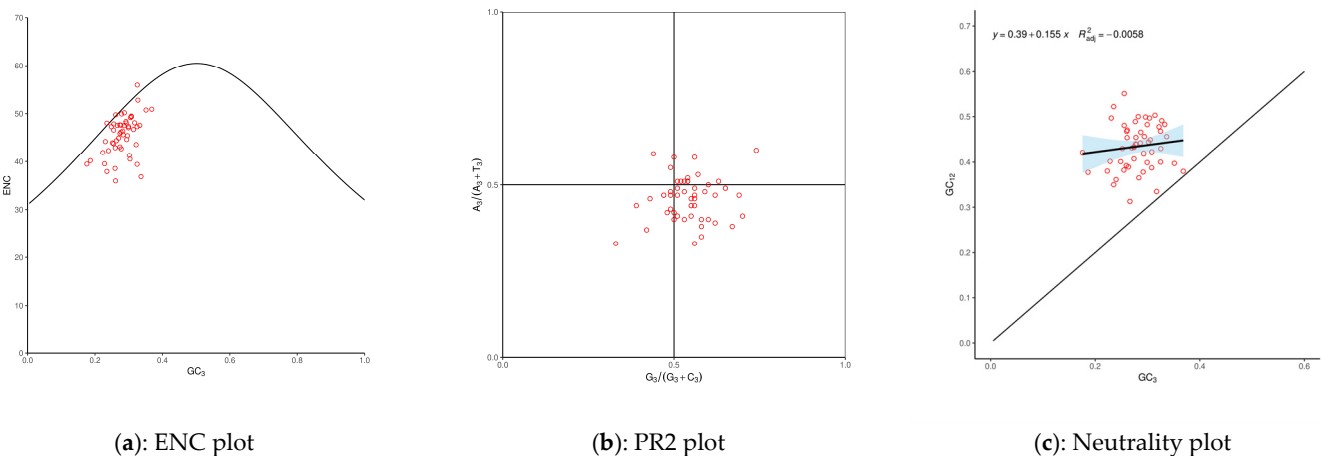

(**a**): ENC plot      (**b**): PR2 plot      (**c**): Neutrality plot

**Figure 3.** Codon usage preference in *D. imperialis* chloroplast genome. ENC: effective number of codons. $GC_{12}$: the average value of $GC_1$ and $GC_2$ for each gene. $GC_3$: the G or C content in the third base of the codon. $A_3/(A_3 + T_3)$: the probability of A when the third base of the codon is A or T. $G_3/(G_3 + C_3)$: the probability of G when the third base of codon is G or C.

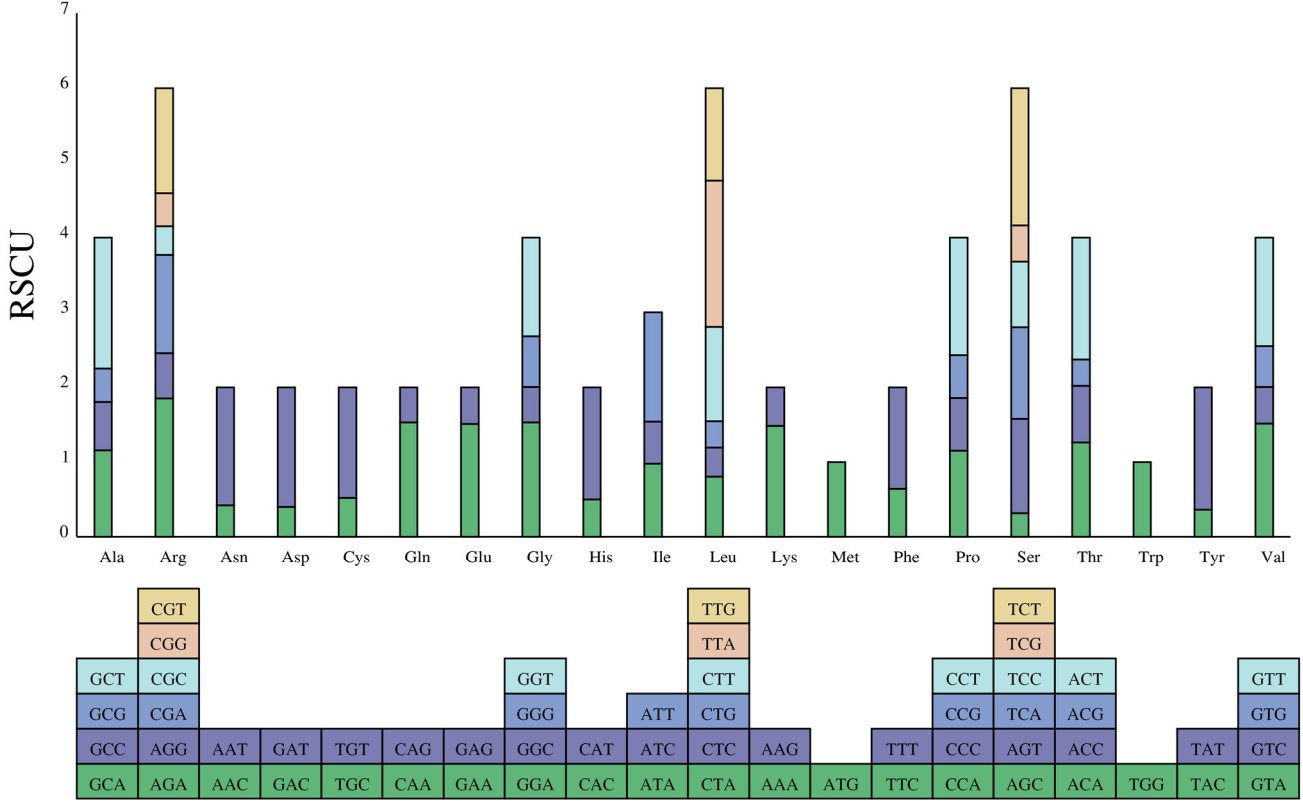

**Figure 4.** The RSCU of amino acids in the *D. imperialis* chloroplast genome. The color of the histogram is the same as the codon color.

### 3.3. Interspersed Repeats and Simple Sequence Repeats (SSRs)

The study identified 62 interspersed repeats in the *D. imperialis* chloroplast genome, including 9 complementary repeats, 6 reverse repeats, 27 palindromic repeats, and 20 forward repeats (Figure 5a). These repetitive sequences were 30 to 25,031 bp long. Except for one repeat that was 25,031 bp long, the other repeats were only 30 to 48 bp long (Table S4).

Further analysis identified a total of 157 SSRs in the *D. imperialis* chloroplast genome, including 133 mononucleotide repeats, 8 dinucleotide repeats, 2 trinucleotide repeats,

12 tetranucleotide repeats, 1 pentanucleotide repeat, and 1 hexanucleotide repeat (Figure 5b). Among these, the most common SSR motif was a mononucleotide repeat containing 8 Ts, with a frequency of 28. The next most common SSR motif was a mononucleotide repeat containing 9 As, with a frequency of 26 (Figure 5c).

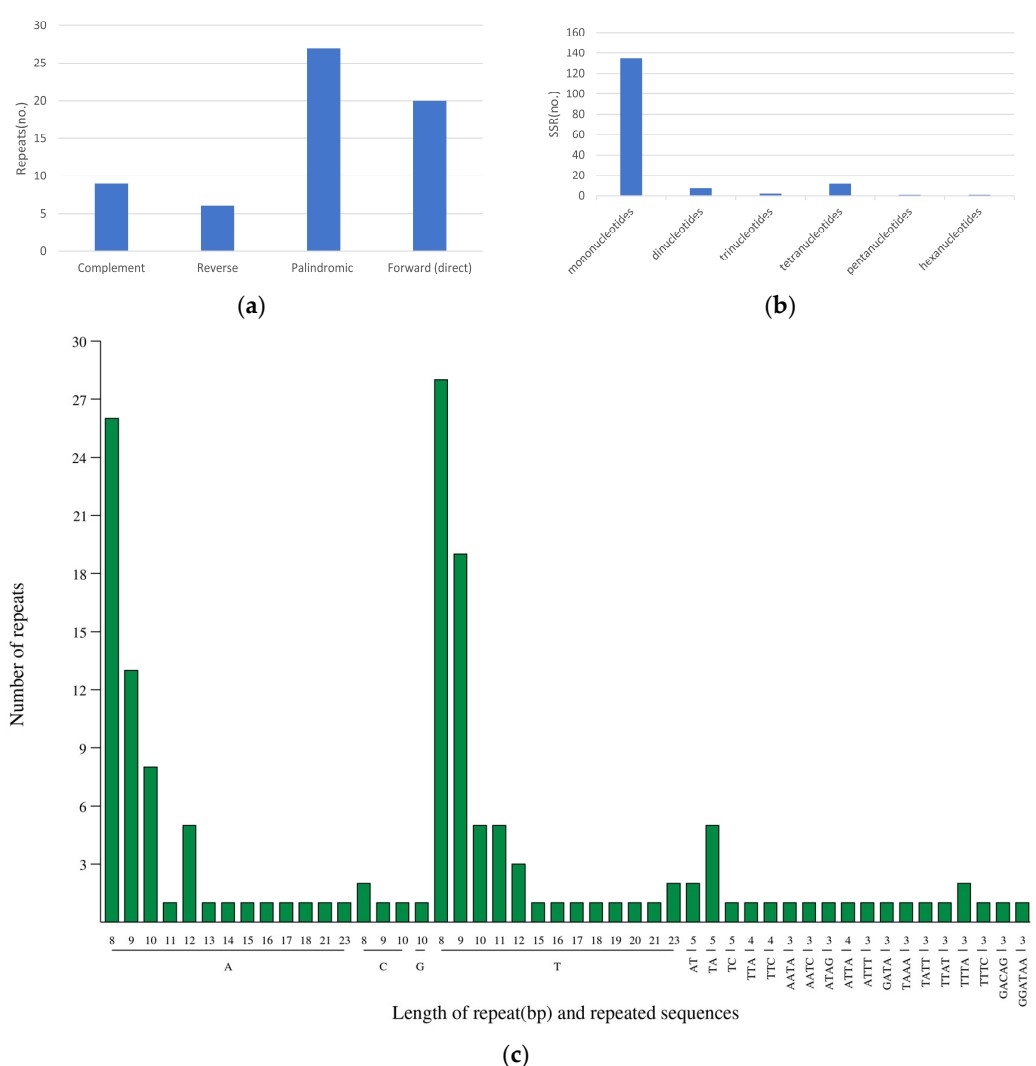

**Figure 5.** SSR types in the chloroplast genome of *D. imperialis*. The horizontal axis represents SSR repeat types, and the vertical axis shows the number of repeats. (**a**) the identification of four types of repeats; (**b**) the type and frequency of each identified SSR; and (**c**) the detail information of all SSR types.

### 3.4. Comparative Analysis of the Chloroplast Genomes of D. imperialis and Its Related Species

Further, to assess the level of variation in DNA sequences of the Heliantheae alliance, we selected 2, 4, and 10 chloroplast genome data sets for sequence comparison among the *Dahlia* species, the Coreopsideae subtribe, and the Heliantheae alliance, respectively. First, we determined the nucleotide diversity (Pi) values of the complete chloroplast genome. The average Pi value was the highest for the SSC region for all three levels. The locus with the highest Pi value in the *Dahlia* genus (Figure 6a) was *psbZ-trnG* (0.03), located in the LSC region. In addition, *psbZ-trnG*, *trnG-trnfM*, *ndhC-trnV* (exon2), *trnL-rpl32*, *ndhF*, and *ndhF-trnN* had high Pi values within the *Dahlia* genus. Within this genus, *psbZ* was identified as a highly variable gene that may be used as a barcode for *D. imperialis*. At the intergeneric level within Coreopsideae (Figure 6b), the locus with the highest Pi value was trnL-rpl32 (0.79) in the SSC region. Additionally, *rpl32-ndhF*, *ycf1*, *rpl32-ndhF*, *trnL-rpl32*, and *accD-psaI* loci had high Pi values at the intergeneric level. Finally, *rpl32*, *ndhF*, *ycf1*, *accD*,

and *psaI* were identified as the highly variable genes among these genera. At the subtribal level, *rpl32-ndhF* (0.07759), located in the SSC region, had the highest Pi value (Figure 6c). Additional loci with high Pi values were *trnL-rpl32*, *ycf1*, *trnL-rpl32*, *rpl32*, *rps16*(exon1)-*trnQ*, *trnC*, and *ycf3*(exon1)-*trnS*. Here, *rpl32*, *ndhF*, *ycf1*, *rps16*, and *ycf3* were identified as highly variable genes, mainly located in the SSC region, with significant mutation.

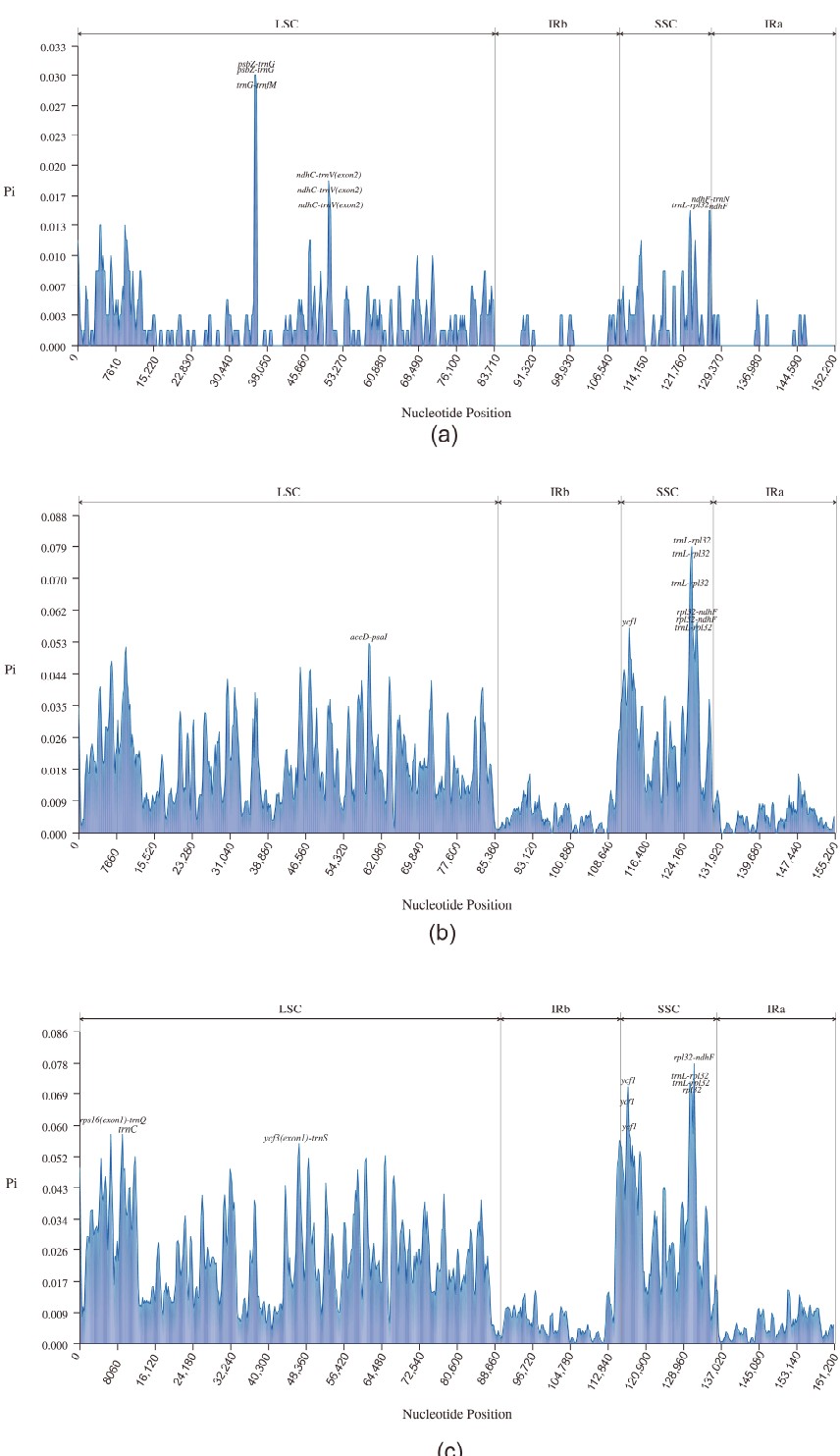

**Figure 6.** Comparative analysis of nucleotide diversity at three levels. The horizontal coordinate indicates the nucleotide position and the vertical coordinate indicates the Pi value (window length: 600 bp; step size: 200 bp). (**a**): the internal species level of *Dahlia*; (**b**): the internal genus level of Coreopsideae; and (**c**): the internal subtribal level of the Heliantheae alliance.

We then analyzed the contraction and expansion of the IR/LSC and IR/SSC junctions across 10 chloroplast genomes. The results showed extremely high similarity within the *Dahlia* genus (Figure 7). Compared with *D. pinnata*, *D. imperialis* had a 3 bp contraction on the SSC side of the IR region. Within Coreopsideae, all four chloroplast genomes were extremely similar. Compared with *Dahlia*, the genera *Bidens* and *Cosmos* had a 30–50 bp shrinkage in the SSC region. Then, we compared the contraction and expansion of the IR/LSC and IR/SSC at the subtribal level. *L. burkei* had noticeably fewer bases than the other nine species (approximately 180 bp) in the LSC side but significantly more bases on the IRb side. This observation suggests a clear expansion from the IR region towards the LSC region in *L. burkei.* In addition, the bases of *E. fortunei* on both sides of the SSC (344 bp and 912 bp) were significantly longer than those in the other nine species, indicating an apparent contraction on the IR side close to *E. fortunei*'s SSC.

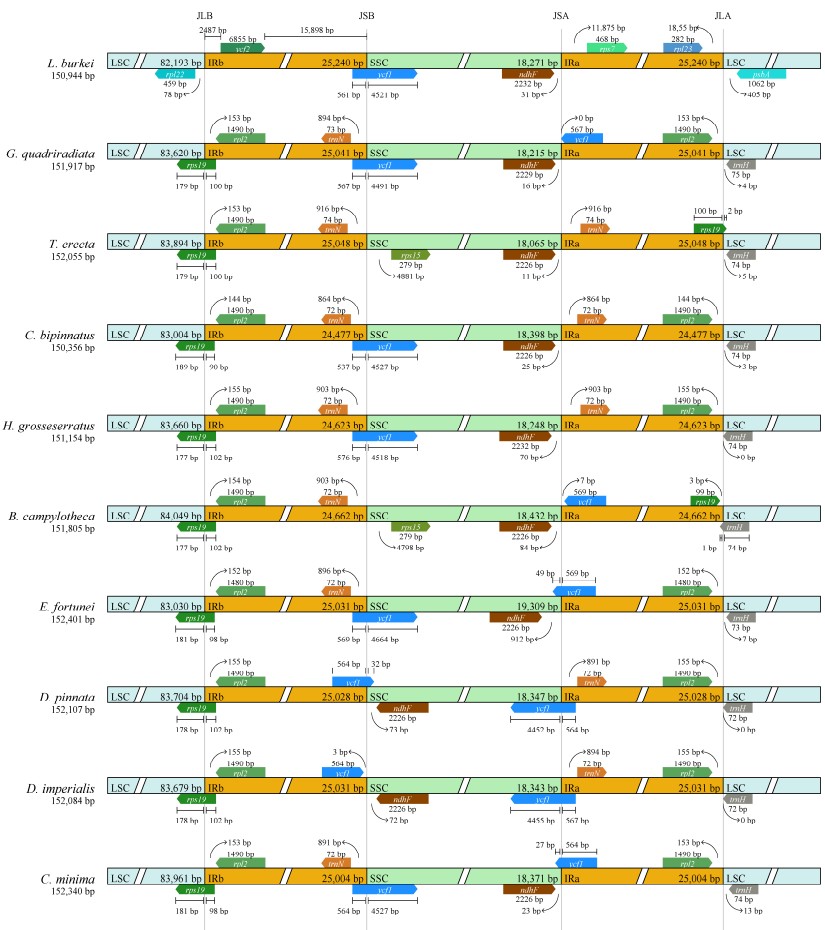

**Figure 7.** Comparison of LSC, SSC, and IR regional boundaries of chloroplast genomes between 10 species. JLB: junction line between LSC and IRb; JSB: junction line between IRb and SSC; JSA: junction line between SSC and IRa; JLA: junction line between IRa and LSC. The start and end of each gene from the junctions is shown with arrows.

Finally, we compared the complete chloroplast genomes of nine representative species from six subtribes in the Heliantheae alliance using *C. minima* as a reference (Figure 8). Analysis of the Ka/Ks ratio of 79 protein-coding genes in these 9 chloroplast genomes showed values less than 1 for most encoded genes; however, a few genes had a Ka/Ks value of zero, indicating the conserved nature of these genes. No genes with a Ka/Ks value greater than 1 were identified among any of the nine species. However, the Ka/Ks value exceeded 1 for *accD* of *D. imperialis* and *D. pinnata*, *rpoB* of *L. burkei*, *rpl33* of *C. bipinnatus*, and *ycf2* of *B. campylotheca*, suggesting the influence of natural selection on the base mutations in these genes.

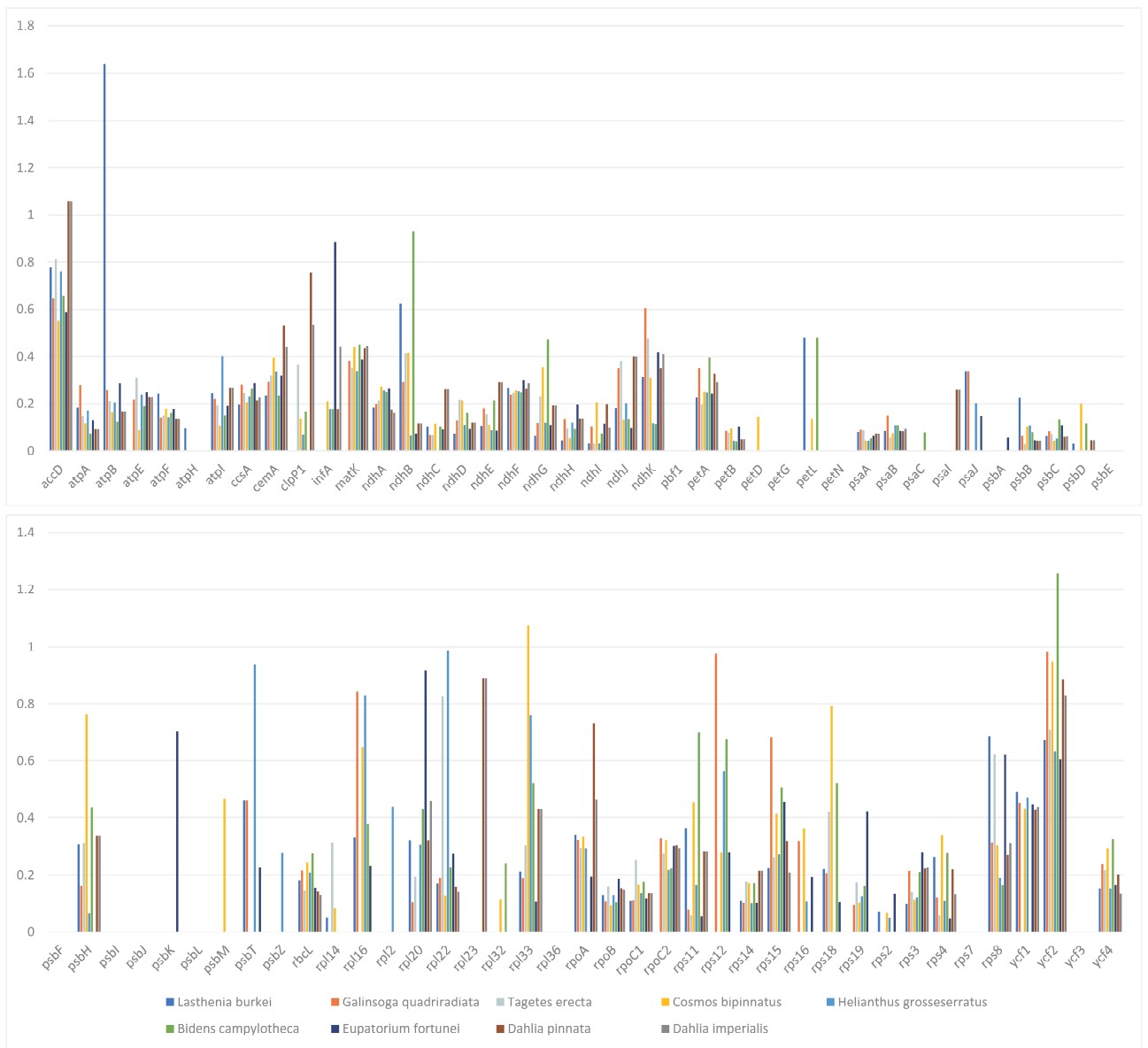

**Figure 8.** The Ka/Ks ratios of 79 protein-coding genes from the chloroplast genomes of *D. imperialis* and nine related species. The same color represents the same species. Blue rectangle: *L. burkei*; orange–red rectangle: *G. quadriradiata*; light grey rectangle: *T. erecta*; orange rectangle: *C. bipinnnatus*; light blue rectangle: *H. grosserratus*; green rectangle: *B. campylotheca*; dark blue rectangle: *E. fortunei*; brown rectangle: *Dahlia pinnata*; dark gray rectangle: *Dahlia imperialis*.

### 3.5. Phylogenetic Inference

In order to determine the position of *D. imperialis* in phylogeny, we downloaded 49 complete chloroplast genomes from NCBI, including 46 chloroplast genomes of the Heliantheae alliance and 3 outgroups, and reconstructed the phylogenetic tree of the Heliantheae alliance. The maximum likelihood tree reconstructed by PhyML had 42 nodes, of which only one node had a relatively low support rate (53%), six nodes had a support rate of 80–99%, and all others had a support rate of 100% (Figure 9). This observation indicated the high reliability of the clustering results. The phylogenetic tree showed that 45 complete chloroplast genomes of 24 genera covered 6 subtribes of the Heliantheae alliance. Here, the *Dahlia* species formed a monophyletic group, with *D. imperialis* located at the group's base. In this study, *Bidens* was positioned as a sister group to *Cosmos*, whereas an earlier nuclear genome analysis indicated that *Dahlia* was the sister group to *Cosmos* [13]. Moreover,

members of the Heliantheae subtribe demonstrated a closer phylogenetic relationship with Eupatieae–Madieae, whereas nuclear genomic topology previously revealed that Heliantheae and Corepsideae were sister groups.

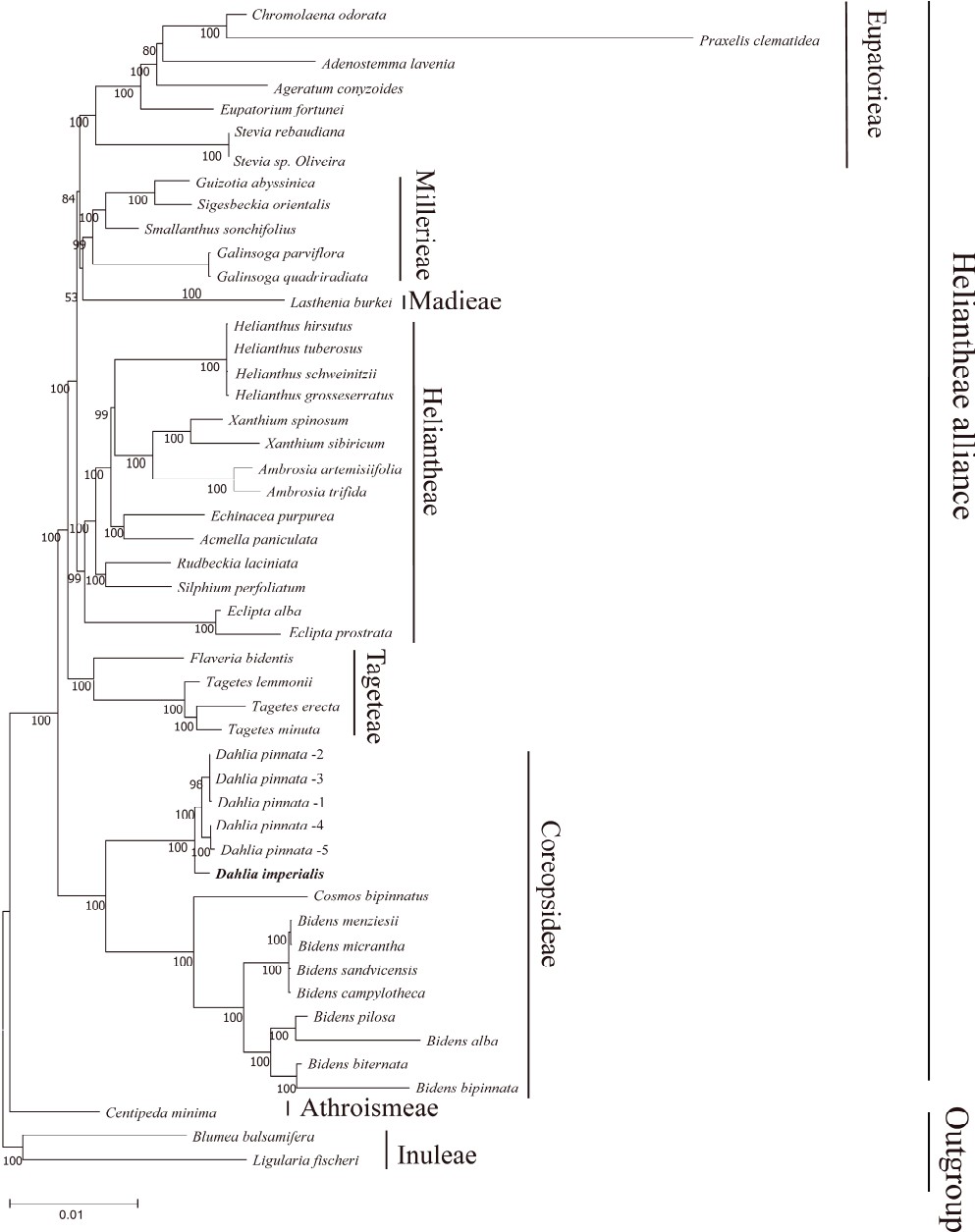

**Figure 9.** Phylogenetic tree based on 49 chloroplast genomes. A phylogenetic tree based on complete chloroplast genomes was generated using maximum likelihood (ML) analysis with 1000 bootstrap replicates. The numbers at each node indicate the bootstrap support values. Five *D.pinnata* chloroplast genomes were distinguished by 1–5.

Further, we conducted network evolution and gene flow analysis on 49 chloroplast genomes to explore the topological structure differences. The network evolution showed a complex network structure, indicating numerous hybridization events within Coreopsideae among *Dahlia*, *Cosmos*, and *Bidens* within the *Dahlia* genera and *Bidens* (Figure 10). Moreover, a few hybridization events were detected within Heliantheae. Subsequent gene flow analysis showed that the arrow color between *Dahlia*, *Cosmos*, and *Bidens* was the reddest, indicating that there was a high degree of gene exchange between the three based on the highest migration weight (Figure 11). Therefore, we speculated that frequent genetic

exchange and hybridization events were the main reasons for the inconsistency in the phylogeny. We also assume that *Cosmos* might have formed due to hybridization between *Dahlia* (the paternal parent) and *Bidens* (the maternal parent).

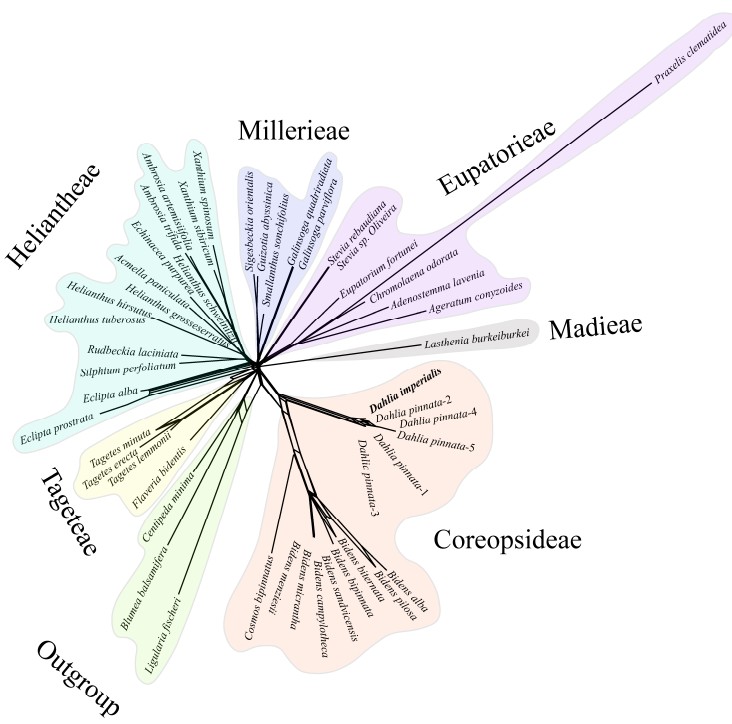

**Figure 10.** Networked evolutionary relationships among 49 chloroplast genomes. The same color module represents the same subtribe. Five *D.pinnata* chloroplast genomes were distinguished by 1–5.

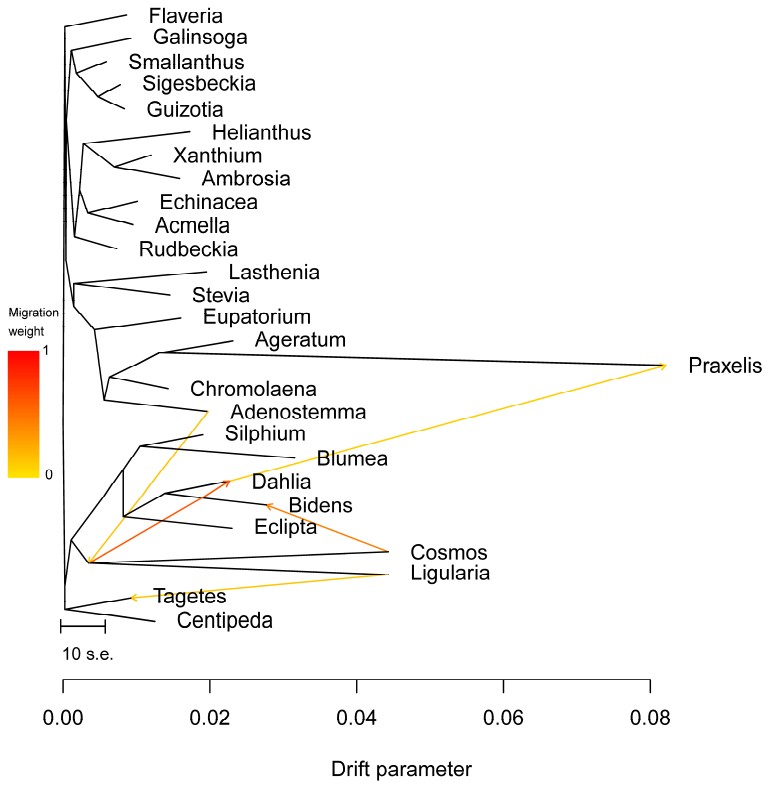

**Figure 11.** Maximum likelihood tree of 49 chloroplast genomes. Populations with inferred migration edges. The arrow represents the infiltration event.

## 4. Discussion

The cultivation history of *Dahlia* is short, and the commercial species have undergone multiple hybridizations and selections during the breeding process [3]. However, there are few species applied. The current commercial species of *Dahlia* is mainly formed through breeding, mutation backcrossing, and reciprocal crossbreeding techniques originating from *D. pinnata* and *D. coccinia* from Mexico and their hybrid offspring [1,4]. Therefore, for the genus *Dahlia* (about 40 species), a large number of species resources are not applied in the production practice of *Dahlia*. A multitude of wild traits and unknown superior genes lie dormant in numerous wild species of *Dahlia*. Effective interpretation, development, and utilization of the superior original species resources of *Dahlia* could bring new vitality to the research and industrial application. The interpretation of the chloroplast genome of *D. imperialis* is a good starting point.

Analyzing the chloroplast genome sequences provides comprehensive information for plant phylogenetic studies. In addition, using chloroplast markers in DNA barcoding helps identify plant species efficiently [42,43]. The *Dahlia* genus has approximately 40 species [1,2], of which only *D. pinnata* has a published complete chloroplast sequence. This limits its systematic evolution research. The present study successfully assembled the complete chloroplast genome of *D. imperialis* using Illumina sequencing data, providing resources for promoting phylogenetic studies.

Like most land plants, *D. imperialis*, the species analyzed in this study, has a chloroplast genome with a highly conserved structure and gene content. Similar to other flowering plants, *D. imperialis* also lacks *chlB*, *chlL*, *chlN*, and *trnP-GGG* genes [44]. Typically, a higher GC content increases DNA density, enhancing the sequence's conservation and rigidity [45]. Our results showed variations in GC content amongst different regions, with the IR region exhibiting a higher GC content primarily due to the presence of rRNA [46]. The high GC content in these regions probably plays a crucial role in maintaining the chloroplast genome's fundamental composition and structural stability.

Interspersed repeats and SSRs are widely present in plant chloroplast genomes [47]. Interspersed repeats are crucial for studying genome recombination and rearrangement and can cause substitutions and insertions in the chloroplast genome [48]. Due to the variations in the type, quantity, and location of repetitive sequences among species, they are widely used for identifying mutation loci and establishing phylogenetic relationships [49]. The present study identified numerous interspersed repeats and SSRs in *D. imperialis* compared to the Senecioneae [50] and Sonchus [51] species of the Asteraceae family, which is advantageous for *Dahlia* phylogenetic research. Moreover, these SSR markers, which are associated with desirable traits, can be utilized in breeding and developing improved target varieties [48].

Furthermore, the highest nucleotide diversity value was detected for the SSC in the chloroplast genome of *D. imperialis*. In addition, *trnL-rpl32-ndhF* was identified as the most highly mutative genetic locus shared at all three levels. In addition to the introns of *trnL* and *ndh F*, which have been used in the phylogeny of Asteraceae [11], we also found relatively high Pi values for *accD*, *psbL*, *ycf1*, *psbZ*, *ycf3*, and *rps16*, which serve as candidate regions for molecular markers at different levels. In addition, the Ka/Ks values of most genes in the six representative subtribe species of the Heliantheae alliance were less than 1, indicating that purifying selection was present among these species. Interestingly, the Ka/Ks values of *accD* in *Dahlia*, *rpl33* in *C. bipinnatus*, and *ycf2* in *B. campylotheca* were greater than 1, which indicates that these genes were affected by environmental selection. In *Arabidopsis*, variations in the editing efficiency of *accD* directly affect the activity of the fatty acid biosynthetic enzyme acetyl-CoA carboxylase (ACCase). This difference influences the synthesis of unsaturated fatty acids and the stability of biological membranes at high temperatures [52]. In *Tobacco*, *rpl33* is essential for plants under cold stress [53]. However, the function of *ycf2* is currently unknown. Therefore, *accD* and *rpl33* are the genes involved in the response to abiotic stress. In addition, the genes with Ka/Ks values greater than 1 were different among the three species, which indicates the diversity

in genetic loci and molecular mechanisms that these three species utilize in response to environmental selection.

Chloroplast genomes have been widely utilized in reconstructing phylogenetic relationships and understanding evolutionary history [54]. The phylogenetic tree reconstructed in this study based on 49 chloroplast genomes is largely consistent with those constructed from 10 chloroplast gene markers [10]. However, some topological differences exist compared to the phylogenetic tree reconstructed based on nuclear genomes [12]. Members of the Heliantheae subtribe demonstrated a closer phylogenetic relationship with Eupatieae–Madieae, whereas nuclear genomic topology previously revealed that Heliantheae and Corepsideae were sister groups, consistent with the findings of Liu et al. [12]. Based on phylogenetic inconsistency, we further performed networked evolution analysis and gene flow analysis. The results showed the occurrence of extensive hybridization events within *Dahlia*, similar to its breeding history with *D. pinnata*. The networked evolution analysis, gene flow analysis, and the differences in the topologies of nuclear and chloroplast genomes among the three genera (*Dahlia*, *Cosmos*, and *Bidens*) further support the significant role played by *Dahlia* in the origin of *Cosmos*. Additionally, we identified two sites of phylogenetic incongruence (putative hybrid origin sites) based on our limited samples. Thus, additional sites of phylogenetic incongruence within the Heliantheae alliance, the broader subfamily of Asteraceae, or even within a wider range of Asteraceae plants are unknown.

## 5. Conclusions

The present study determined the chloroplast genome sequence and structure of *D. imperialis*, which are highly similar to those of the Heliantheae alliance. The comprehensive analysis of the complete chloroplast genome of *D. imperialis* enriches the information we have on the genetic resources of *Dahlia* species. The work also detected codon usage preference in *D. imperialis*, which probably resulted from natural selection and mutations. We also identified numerous repeat sequences and SSRs, which could serve as potential sources for molecular markers. Comparative analysis of the chloroplast genomes revealed highly variable genes at different levels and differences in genes involved in environmental adaptation among species. The results of the phylogenetic, network evolution, and gene flow analysis indicate that extensive hybridization events have occurred within *Dahlia*, consistent with the artificial breeding history of this genus. It is necessary to obtain more chloroplast genome information of the *Dahlia* germplasm or Asteraceae species to further improve the phylogenetic relationship of *Dahlia* and Asteraceae.

**Supplementary Materials:** The following supporting information can be downloaded at: https://www.mdpi.com/article/10.3390/horticulturae10010007/s1, Figure S1: Read coverage depth map in *D. imperialis* chloroplast genome; Table S1: Reconstruction of the sequence information of the phylogenetic tree; Table S2: Gene composition in this chloroplast genome; Table S3: Codon usage in the *D. imperialis* chloroplast genome; Table S4: Interspersed repeat information in the *D. imperialis* chloroplast genome.

**Author Contributions:** Conceptualization, S.-C.N. and D.-F.C.; methodology, S.-C.N.; software, S.-D.D.; validation, Y.L., L.-H.H. and D.-Y.X.; formal analysis, S.-D.D.; investigation, D.-Y.X. and W.-B.Y.; resources, J.L. and W.-B.Y.; data curation, S.-C.N. and L.-H.H.; writing—original draft preparation, S.-D.D.; writing—review and editing, S.-C.N., S.-D.D., D.-F.C., L.-H.H. and Y.L.; visualization, Y.L.; supervision, S.-C.N., D.-F.C., J.L. and W.-B.Y.; project administration, S.-C.N. and D.-F.C.; funding acquisition, S.-C.N. and D.-F.C. All authors have read and agreed to the published version of the manuscript.

**Funding:** This work was supported by the Hebei Provincial Key R&D Program Project (20326814D), the Hebei Provincial Colleges and Universities Basic Research Task Fee Research Project (KY2021055), and the Natural Science Fund Project of Hebei Province (C2022204240).

**Data Availability Statement:** The genome sequence data that support the findings of this study are openly available in GenBank of NCBI at [https://www.ncbi.nlm.nih.gov] under the accession no. OP323060. The associated BioProject, SRA, and Bio-Sample numbers are PRJNA887522, SRR22094635, and SAMN31170944 respectively.

**Conflicts of Interest:** The authors declare no conflict of interest.

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
