# Peer review of "Complete Chloroplast Genome Sequence of Dahlia imperialis (Asteraceae): Comparative Analysis and Phylogenetic Relationships"

_horticulturae, doi:10.3390/horticulturae10010007_

Round 1

Reviewer 1 Report

Comments and Suggestions for Authors

Review report

The manuscript in title: Complete chloroplast genome sequence of Dahlia imperialis 2 (Asteraceae): comparative analysis and phylogenetic relationships. By Duan et al., describes the complete chloroplast genome of D. imperialis for the first time.

The same author(s) did similar work previously with sequencing the chloroplast genome of Dahlia pinnata which is under publication elsewhere, reflecting a trend by the author(s). The manuscript is well design and written but there are some points need to be resolved before further consideration.

L33-34: change (It is native to Mexico and is the national flower of Mexico) to (It’s native and the national flower of Mexico).

L38: Delete “and” from “period and large” and replace with ”,”

L54: change “in South America and North America” to “in South and North America”

L80-84: this section is a method section, it should be transferred to the materials and methods sections.

L88-90: Authors say that the “The interpretation of chloroplast genome information of D. imperialis enriched the genetic information of Dahlia and laid………etc”, How this can happen? Your manuscript still under evaluation and not accepted yet, how this information enriched the genetic information of Dahlia and laid a foundation for the innovation (this is a past tense)? You have to change to the future tense.

L97: delete “and” from “with 75% alcohol, and then rinsed”.

L98: put the sentence in past simple tense and delete “were” from “and were stored at-80 °C after”

L99: How authors extracted DNA from leaves, no methods or references have been described?

L99: In the DNA extraction and genome sequencing section, Authors should add some details about the instruments that they used such as The 2100 Bioanalyzer, qPCR (the model and company of these instruments).

L121: change the letter ”c” to capital form.

L131: change “the ENC plot, a two-dimensional scatter plot, was generated” to “the ENC plot and a two-dimensional scatter plot, were generated”.

L146:147: including those of the two Dahlia species for comparing sequence variation within the genus Dahlia, and two Dahlia? Dahlia is mentioned twice, How?

L149: In addition, a representative species was selected, change “was” to “were”.

L169: The complete genome sequence was 152,1084 bp, How can you read this number? I think its 152,084 bp as mentioned in the abstract.

L221: followed by isoleucine (1,795, 8.51%), does this correct? From fig 4, its expected to be Arginine?

L223: The most preferred codon was UUA, which encoded leucine (Leu)? Do you mean TTA codon? UUA is not exist in fig 4.

L290-291: No genes with a Ka/Ks value all greater than 1? Re-write to clarify the meaning.

L310: Besides, not italic

L343: delete “of Dahlia”

L361-362: re-write as follows: Interspersed repeats are crucial for studying genome recombination, rearrangement, in addition to cause substitutions and insertions in the chloroplast genome

L371: Besides, not italic

L384-387: Re-write to clarify the meaning.

L410: Dahlia should be italic

L416-417: that extensive hybridization events have occurred within Dahlia,…………. of the genus Dahlia. Repeating Dahlia here is not a good writing method, delete “of the genus Dahlia” and replace with “of this genus”.

Comments on the Quality of English Language

Reviewer 2 Report

Comments and Suggestions for Authors

In this manuscript, Duan et al. present a comprehensive analysis of the complete chloroplast genome of Dahlia imperialis, providing a valuable genomic resource for future studies on Dahlia plants. The manuscript is commendably organized and well-presented; however, attention to detail is crucial for enhancing its overall quality.

1. In Figure 1, panels (a) and (b), ensure that gene names are appropriately italicized for clarity.

2. Figure 2, Line 200, lacks a figure legend. Similar issues are observed in Figures 3, 8, and 10. Authors are advised to provide detailed figure legends for better comprehension.

3. Lines 201-208: Standardize the format of figure citations, such as changing "Figure 3-a" to "Figure 3a." Consistency in formatting should be maintained throughout the manuscript.

4. Line 209: Clarify the reference of "the two" in the statement, "The R2 value of 0.0058 indicates no significant correlation between the two."

5. Lines 223 and 224: If abbreviations for leucine and arginine are intended, introduce them when these amino acids are first mentioned.

6. Line 261: Italicize "rpl32" for proper gene name formatting.

7. Figure 7: Improve the resolution to enhance text visibility, ensuring clarity in the presented information.

8. Lines 292-293: Italicize gene names for consistency and readability.

9. Line 301: Ensure that unnecessary italics are removed.

10. Line 354: Italicize gene names in this context for consistency with the manuscript style.

Reviewer 3 Report

Comments and Suggestions for Authors

The manuscript titled "Complete chloroplast genome sequence of Dahlia imperialis (Asteraceae): comparative analysis and phylogenetic relationships" by Shan-De Duan et al. presents a comprehensive study on the chloroplast genome of Dahlia imperialis, a species within the Dahlia genus. The authors have sequenced, assembled, and annotated the complete chloroplast genome of D. imperialis for the first time, providing valuable insights into the phylogenetic relationships within the Dahlia genus and the Heliantheae alliance. The manuscript is well-structured and provides a detailed account of the methods used for DNA extraction, genome sequencing, annotation, codon usage analysis, interspersed repeats and SSRs, comparison of the complete chloroplast genomes, and phylogenetic analysis. The authors have also provided a comprehensive introduction to the Dahlia genus and the importance of studying its chloroplast genome. The chloroplast genome of D. imperialis was found to be 152,084 bp long with a typical quadripartite structure and a GC content of 38.45%. A total of 134 genes were annotated in the genome, including 86 protein-coding genes, 38 transfer RNA genes, 8 ribosomal RNA genes, and 2 pseudogenes. The detailed analysis identified UUA as the most frequently used codon and found 62 intergenic repeat sequences and 157 SSR loci in the D. imperialis genome. The phylogenetic analysis based on 49 chloroplast genomes showed that Dahlia was a monophyletic group, with D. imperialis positioned at the base of this clade. Network evolution and gene flow analysis unveiled extensive hybridization events within the Heliantheae alliance, especially in Dahlia. The manuscript is well-written and provides a significant contribution to the field of plant genomics. The findings of this study enrich the information of genetic resources of Dahlia species, provide valuable information for reconstructing their phylogeny, and reveal the evolutionary dynamics of the Dahlia chloroplast genome. However, the manuscript could benefit from a more detailed discussion on the implications of these findings for the conservation and utilization of Dahlia species. Additionally, the authors could consider including more visual aids, such as charts or diagrams, to help readers better understand the complex genetic and phylogenetic analyses.

Specific comments:

  1. Table 2 can be moved to supplementary file, as the gene content is similar to previously sequenced Dahlia plastomes

  2. Figure 6 - this figure have to be improved, I would recommend using some circular visualization, like circos, and plot value line using different colors for each taxon

  3. Figure 7 - Please remove species with similar structures

  4. Figure 9 - this figure has to be improved in the branch length - it should be far more resolved, especially when there's room for it. Names of the families can be rotated vertically to improve general layout.
